# Two-magnon scattering in the 5$d$ all-in-all-out pyrochlore magnet Cd$_2$Os$_2$O$_7$

Thi Minh Hien Nguyen[1,2], Luke J. Sandilands[1,2,9], C.H. Sohn[1,2], C.H. Kim [1,2], Aleksander L. Wysocki[3], In-Sang Yang[4], S.J. Moon[5], Jae-Hyeon Ko[6], J. Yamaura[7], Z. Hiroi[8] & Tae Won Noh[1,2]

5$d$ pyrochlore oxides with all-in-all-out magnetic order are prime candidates for realizing strongly correlated, topological phases of matter. Despite significant effort, a full understanding of all-in-all-out magnetism remains elusive as the associated magnetic excitations have proven difficult to access with conventional techniques. Here we report a Raman spectroscopy study of spin dynamics in the all-in-all-out magnetic state of the 5$d$ pyrochlore Cd$_2$Os$_2$O$_7$. Through a comparison between the two-magnon scattering and spin-wave theory, we confirm the large single ion anisotropy in this material and show that the Dzyaloshinskii–Moriya and exchange interactions play a significant role in the spin-wave dispersions. The Raman data also reveal complex spin–charge–lattice coupling and indicate that the metal–insulator transition in Cd$_2$Os$_2$O$_7$ is Lifshitz-type. Our work establishes Raman scattering as a simple and powerful method for exploring the spin dynamics in 5$d$ pyrochlore magnets.

[1] Center for Correlated Electron Systems, Institute for Basic Science (IBS), Seoul 151-742, Republic of Korea. [2] Department of Physics and Astronomy, Seoul National University (SNU), Seoul 151-742, Republic of Korea. [3] Ames Laboratory, U.S. Department of Energy, Ames, Iowa 50011, USA. [4] Department of Physics and Division of Nano-Sciences, Ewha Womans University, Seoul 03760, Republic of Korea. [5] Department of Physics, Hanyang University, Seoul 04763, Republic of Korea. [6] Department of Physics, Hallym University, Chuncheon, Gangwondo 24252, Republic of Korea. [7] Materials Research Center for Element Strategy, Tokyo Institute of Technology, Kanagawa 226-8503, Japan. [8] Institute for Solid State Physics, University of Tokyo, Kashiwa 277-8581, Japan. [9] Present address: Measurement Science and Standards, National Research Council of Canada, Ottawa, Ontario, Canada K1A 0R6. Correspondence and requests for materials should be addressed to T.W.N. (email: twnoh@snu.ac.kr)

Pyrochlore lattice magnets feature a network of corner-sharing tetrahedra, whose triangle-based lattice geometries can result in geometrically frustrated magnetism[1]. A number of $5d$ transition metal-oxide pyrochlores, including $Cd_2Os_2O_7$[2–7] and $R_2Ir_2O_7$ ($R$ = Sm[8], Nd[9], Y[10], Eu[11], etc) are believed to host all-in-all-out (AIAO) antiferromagnetic ground states. A schematic of the AIAO ordered state in $Cd_2Os_2O_7$ is shown in Fig. 1b. The Os spins in a given Os-O tetrahedron point inward (all-in), while those in the neighboring tetrahedra point outward (all-out). In the $5d$ transition metal-oxide pyrochlores, AIAO order is often accompanied by a continuous metal–insulator transition (MIT). The close relationship between AIAO order and electronic structure is especially intriguing since AIAO order breaks time-reversal symmetry while retaining the crystalline symmetry of the pyrochlore lattice. Based on such symmetry breaking, numerous theoretical works have proposed that these systems may realize strongly correlated and topologically nontrivial states, in particular the Weyl semimetallic state[12, 13]. These proposals have in turn stimulated a large experimental effort[5, 8, 14].

The pyrochlore osmate $Cd_2Os_2O_7$ is a prototype system in which to investigate AIAO magnetism. The AIAO magnetic ordering at the Os sites has been firmly established by resonant X-ray and neutron diffraction below $T_N$ = 227 K[4, 5], and the material is available in high-quality single crystalline form[4]. The AIAO order drives an MIT which has recently been classified as Lifshitz-type[4, 6, 10]. Due to the $5d^3$ ($t_{2g}^3$) configuration, the $Os^{5+}$ ion is in a $L$ = 0, $S$ = 3/2 state[5, 7]. Since this state is an orbital singlet, spin-orbit coupling (SOC) does not, to first order, play a dominant role. Instead, the strong SOC manifests itself as a large single-ion anisotropy (SIA), which is in turn thought to stabilize the AIAO state[2, 7, 15]. This is in contrast with the Ir-based pyrochlores where the SOC leads to the formation of relativistic $j_{eff}$ = ½ orbitals and the AIAO order depends crucially on the antisymmetric Dzyaloshinskii–Moriya interaction (DMI)[16]. The $Cd^{2+}$ ion is non-magnetic, in contrast to most Ir-based pyrochlores $R_2Ir_2O_7$[8, 9], where the rare-earth ions may also order magnetically. These two pyrochlore sublattices (Ir and $R$) may be magnetically coupled, which complicates investigations of the Ir-site AIAO magnetism.

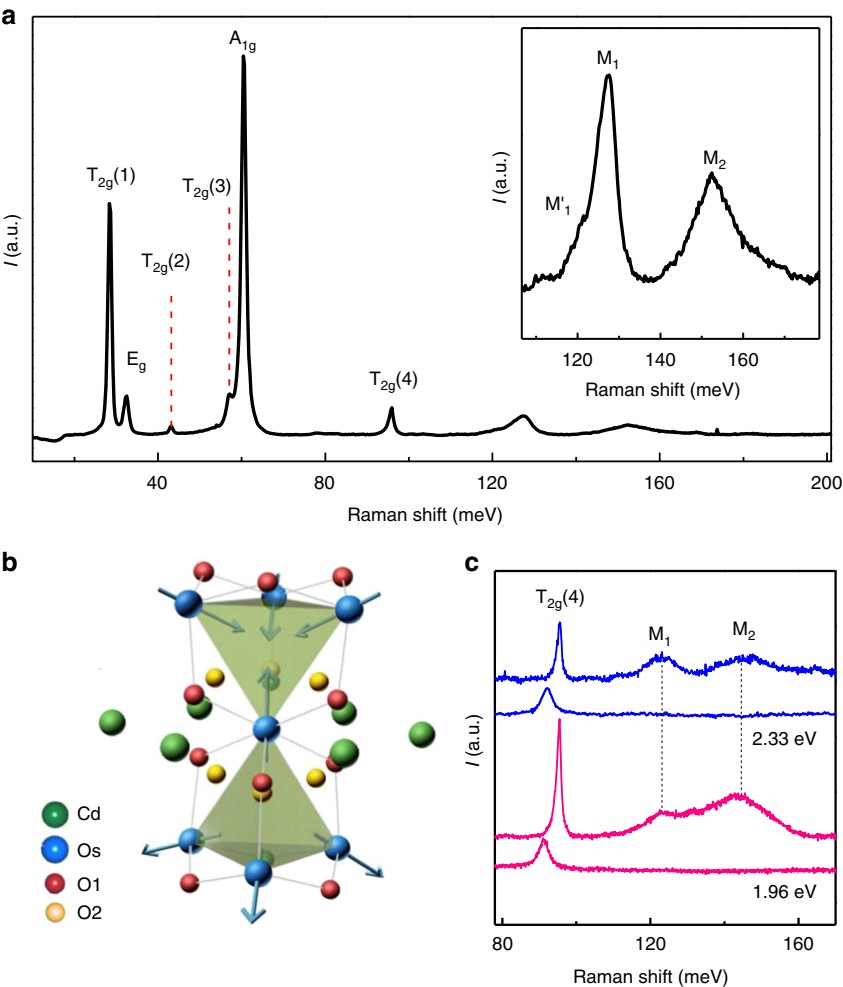

**Fig. 1** Raman spectra of $Cd_2Os_2O_7$ obtained at low temperature. **a** is the Raman spectrum obtained at 15 K under 2.33 eV excitation laser. The spectrum shows six Raman active Γ-point phonon modes with symmetries $A_{1g}$ + $E_g$ + $4T_{2g}$. *Inset* of **a** is a detail showing the broad peaks located in the frequency region between 105–175 meV. Three inelastic features are resolvable: $M_1$ at ~127 meV, $M_2$ at ~152 meV and the shoulder $M_1'$ at ~121 meV. **b** is the pyrochlore lattice structure showing the Os (*blue*) tetrahedra with AIAO magnetic ordering (*blue arrows*). In AIAO magnetic order, the transition metal spins belonging to a given tetrahedron either all point directly towards or away from the tetrahedron's center. The Cd ions are in *green*. Trigonally compressed cages of oxygens O(1) (*red*) surround each Os, and the oxygens O(2) (*yellow*) are bonded to Cd. **c** is the Raman spectra obtained at 100 K and 240 K under 2.33 eV and 1.96 eV excitations. The *blue spectra* and *pink spectra* were obtained under 2.33 eV and 1.96 eV, respectively. For each excitation, the top spectra were measured at 100 K and the bottom spectra were measured at 240 K

While a number of studies have investigated the magnetic excitations in 5$d$ pyrochlores, a full understanding of the AIAO magnetism remains elusive. This is partly due to the fact that the magnetic excitations in these systems have proven difficult to study with conventional techniques such as inelastic neutron scattering or resonant inelastic X-ray scattering (RIXS). Cd and Ir are strong neutron absorbers[17, 18], making inelastic neutron scattering challenging. Several RIXS studies have examined the spin excitations in AIAO magnets. Donnerer et al.[8] recently reported evidence for a dispersive spin-wave excitation in Sm$_2$Ir$_2$O$_7$, while Clancy et al.[14] found evidence for a dispersive and heavily damped magnetic excitation in Eu$_2$Ir$_2$O$_7$ and Pr$_2$Ir$_2$O$_7$. However, the resolution of state-of-the-art RIXS is still only 25 meV for Ir[8] and above 40 meV for Os[14], so fine details of the magnetic excitation spectrum, including the dispersions and energy splitting between distinct magnetic excitations are difficult to resolve. In the case of Cd$_2$Os$_2$O$_7$, a recent RIXS experiment observed a dispersionless magnetic excitation that was interpreted as a superposition of single-site spin-flip excitations[15]. The magnetic excitation in Cd$_2$Os$_2$O$_7$ was found to be broad and featureless, so a detailed determination of the spin Hamiltonian from the experimental data was not attempted. As a result, the nature of the magnetic excitations of the AIAO state in Cd$_2$Os$_2$O$_7$, their relationship with other degrees of freedom, and a quantitative understanding of the underlying microscopic spin Hamiltonian have not been clarified to date.

In this article, we report a combined Raman spectroscopic and theoretical investigation of the magnetic excitations in Cd$_2$Os$_2$O$_7$. The fine energy resolution (~0.25 meV) of the Raman technique[19] allows us to observe two well-separated two-magnon peaks that were not detected with other spectroscopies. Our results constitute the first detection of two-magnon excitations in a 5$d$ pyrochlore oxide using Raman scattering. Through a comparison with linear spin-wave theory (LSWT), we make quantitative estimates of the magnetic couplings that are in good agreement with recent many-body quantum-chemical calculations[7] and clarify the impact of the DMI, SIA, and Heisenberg-exchange interaction on the magnetic excitation spectrum. Understanding the nature of the magnetic excitations and the role of the microscopic magnetic interactions provides a foundation for deeper and fundamental discussion regarding the magnetic properties. Our measurements also reveal signatures of non-trivial coupling between spin, charge and lattice degrees of freedom across the magnetically-driven MIT which indicates that this transition may be classified as Lifshitz-type. These results provide insight into the interplay between electronic structure and magnetism in 5$d$ pyrochlore magnets and highlight the utility of Raman scattering for exploring the spin dynamics of other 5$d$ AIAO pyrochlore magnets, such as the closely related pyrochlore iridates.

## Results

**Two-magnon Raman scattering**. Below 100 meV, the Raman spectra of Cd$_2$Os$_2$O$_7$ show six phonon modes (see Fig. 1a). The spectra of Fig. 1a were obtained at 15 K using a 2.33 eV excitation laser in an unpolarized configuration. Six phonon modes may be discerned around 28, 32, 43, 57, 60, and 96 meV. These modes are consistent with a factor group analysis, which predicts six Raman-active Γ-point phonon modes with A$_{1g}$ + E$_g$ + 4T$_{2g}$ symmetries for the cubic pyrochlore lattice with space group $Fd3m$[20]. By comparison with previous reports[20, 21], the four peaks at 28, 43, 57, and 96 meV can be assigned to T$_{2g}$, the peak at 32 meV to E$_g$, and the peak at 60 meV to A$_{1g}$. No change in the number of phonon modes was observed between 100 and 270 K, confirming the absence of a structural transition across $T_N$. Further

discussion of the phonon spectra and a comparison with first-principles calculations may be found in the Supplementary Note 1 and Supplementary Table 1.

Above 100 meV, the Raman spectra in Fig. 1 reveal two intriguing broad peaks. These are located at ~127 meV and ~152 meV and marked as M$_1$ and M$_2$, respectively. The M$_1$ peak also possesses a low-energy shoulder, which we label M$_1$'. These features are shown in more detail in the inset of Fig. 1. The widths and energies of M$_1$ and M$_2$ are much larger than the optical phonon modes located below 100 meV. The polarization properties of M$_1$ and M$_2$ were investigated in three input-output polarization geometries: collinear, crossed, and unpolarized. We found that the lineshape and Raman shift of M$_1$ and M$_2$ are independent of polarization geometry (Supplementary Fig. 2). For the remainder of our discussion, we focus on the unpolarized case, as the overall intensity is larger in this geometry. We also measured the Raman scattered intensity under 1.96 eV excitation at 100 K and verified that the Raman shifts of M$_1$ and M$_2$ are unchanged (Fig. 1c). This confirms that M$_1$ and M$_2$ are true Raman features and not from photoluminescence.

In order to elucidate the nature of M$_1$ and M$_2$, we measured the Raman scattered signal at various temperatures ($T$). A 1.96 eV excitation was used for the $T$-dependent measurements because it produces stronger intensity of the M$_1$ and M$_2$ peaks. The results are presented in Fig. 2a, where the spectra are displayed with a vertical offset for clarity. We find that M$_1$ and M$_2$ are strongly $T$-dependent compared to the narrow phonon peaks, suggesting a different origin of these features. As $T$ increases, M$_1$ and M$_2$ rapidly lose intensity and vanish near $T_N$. In addition, both features soften and broaden considerably as $T$ increases. These trends may be better appreciated by considering the difference spectra shown in Fig. 2b. Here we show $I(\omega,T) - I(\omega, 230$ K$)$ with the T$_{2g}$(4) phonon subtracted. The reference temperature of 230 K is just above $T_N$. As $T$ increases, the M$_1$ and M$_2$ peaks become indistinguishable while the overall distribution of scattered intensity broadens and shifts to lower energy. Above $T_N$, no $T$-dependent scattering is observed. This temperature dependence indicates that M$_1$ and M$_2$ have a magnetic origin.

To confirm the connection with magnetism, we compare the Raman intensity with the magnetic susceptibility $\chi$. In Fig. 2c, we show the integrated intensity of the $I(\omega,T) - I(\omega, 230$ K$)$ spectra in the energy region between 75 and 165 meV (*open black circles*) and compare these with the literature values of $\chi$ (*filled blue triangles*)[4]. Note that both quantities show clear anomalies at $T_N$. The integrated intensity is relatively constant below 0.6 $T_N$ before rapidly dropping to a minimum at $T_N$, while $\chi$ peaks at $T_N$. Above $T_N$, the magnetic scattering is negligible (we only observe a temperature-independent, flat background) and the susceptibility is slowly varying. Figure 2d displays the $T$-dependence of the normalized frequency ($\omega_M$), intensity ($I_M$) and linewidth ($\Gamma_M$) of the M$_2$ peak which was fitted with a Lorentzian function. Above 180 K, M$_2$ becomes too broad and weak to analyze in this way. The abrupt decrease in intensity of M$_1$ and M$_2$, as well as the rapid changes in the lineshapes and peak locations (Fig. 2a and Fig. 2d), suggests that these features have a magnetic origin. A recent RIXS study of Cd$_2$Os$_2$O$_7$ also identified a feature near 160 meV of magnetic origin that emerges near $T_N$, similar to our findings[5]. Furthermore, the $T$-dependent Raman scattering of AIAO-ordered Cd$_2$Os$_2$O$_7$ is similar to the two-magnon scattering reported in a number of well-studied magnetic systems[22–26]. These observations suggest that the M$_1$ and M$_2$ features originate from two-magnon Raman scattering.

**Comparison with linear spin-wave theory**. Two-magnon Raman scattering occurs via the excitation of two magnons with total

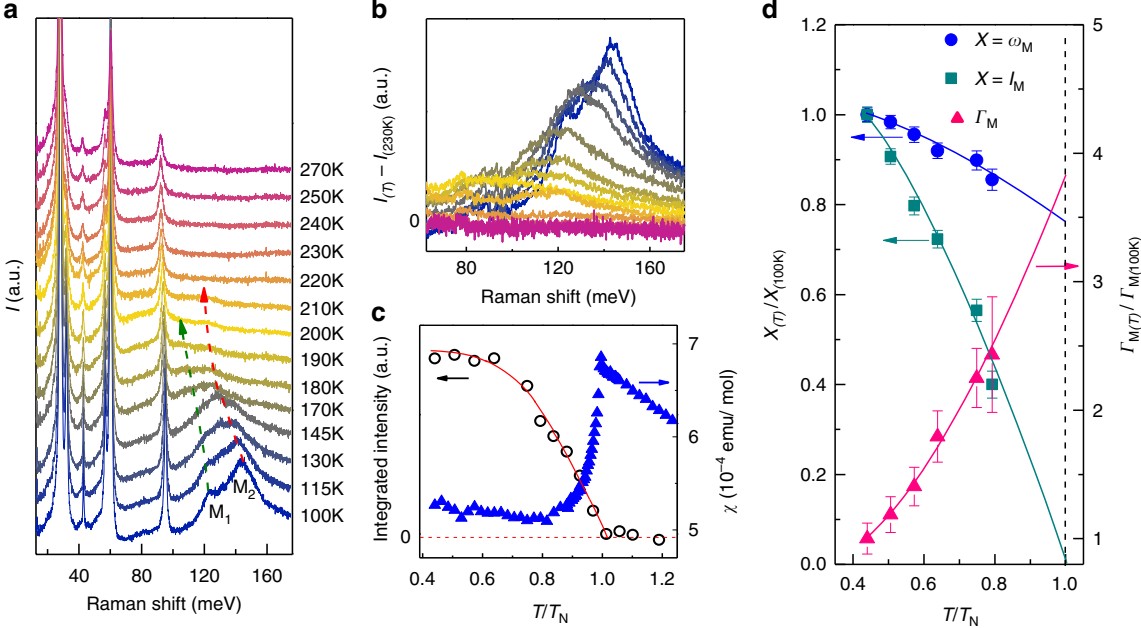

**Fig. 2** $T$-dependent Raman spectra of Cd$_2$Os$_2$O$_7$. **a, b** Raman intensities ($I$) and difference spectra ($I_T$-$I_{230K}$) of Cd$_2$Os$_2$O$_7$ at various temperatures from 100 K to 270 K. The intensity difference spectra were obtained by first removing the T$_{2g}$(4) phonon contribution and then taking the difference between the Raman intensity at a particular temperature and 230 K (just above $T_N = 227$ K). The *dashed arrows* are guides for the eye and represent the evolution of the two-magnon features with $T$. Note that the two-magnon peaks M$_1$ and M$_2$ show a much stronger $T$-dependence than the phonon peaks. **c** $T$-dependence of the integrated intensity of the two-magnon peaks in the range of 75–165 meV (*open black circles*) and the magnetic susceptibility at $H = 20$ kOe measured after zero-field cooling (*filled blue triangles*). The *red solid line* is a guide for the eye. Both quantities show clear anomalies at $T_N$. **d** $T$-dependence of the normalized frequency ($\omega_M$—*filled blue circles*), intensity ($I_M$—*filled dark cyan squares*) and FWHM ($\Gamma_M$—*filled pink triangles*) of the M$_2$ peak. These parameters were derived by fitting M$_2$ with a Lorentzian lineshape. The *error bars* represent the standard deviation in the data fitting procedure. The *solid lines* are guides for the eye

momentum $q = 0$[27]. Ignoring magnon–magnon interaction effects, the scattered intensity is therefore related, within a matrix element, to the two-magnon density of states (DOS). To quantitatively connect the observed magnetic signal with the spin wave dispersion, we computed the magnon DOS within LSWT. The magnon dispersions were calculated for the AIAO magnetic ground state, assuming the following Hamiltonian:

$$\hat{H} = J \sum_{ij}^{nn} \mathbf{S}_i \cdot \mathbf{S}_j + \sum_{ij}^{nn} \mathbf{D}_{ij} \cdot \left( \mathbf{S}_i \times \mathbf{S}_j \right) + A \sum_i \left( \mathbf{S}_i \cdot \mathbf{n}_i \right)^2. \quad (1)$$

The terms are the nearest-neighbor exchange coupling ($J$), the nearest-neighbor DMI ($D$), and the SIA ($A$). $\mathbf{S}_i$ is a spin operator at site $i$ with $S = 3/2$. For $J$, $D$, and $A$, we used values of 5.1 meV, 1.7 meV, and −5.3 meV, respectively. As discussed in detail later, these values yield good agreement with the Raman data and also closely agree with the results of many-body quantum-chemical calculations[7]. The sign of the DM vector was assumed to be of the direct type ($D > 0$) since it stabilizes the AIAO state[7, 8]. The spin easy axis is denoted by a unit vector $\mathbf{n}_i$, which points along the line connecting the centers of the two tetrahedra that adjoin site $i$.

The LSWT calculations confirm that the broad Raman peaks arise from two-magnon Raman scattering in the AIAO state. As displayed in Fig. 3a, our model calculation shows three spin-wave branches along the high-symmetry Γ-X line. From the dispersions, we evaluated the magnon DOS, displayed in Fig. 3b. Note that the DOS is usually dominated by states near the Brillioun zone boundary. Figure 3c shows a comparison between the energy of the M$_1$ and M$_2$ Raman peaks and the magnon DOS with energy scaled by factor of two, which approximates the two-magnon DOS. We find excellent agreement between the

experimental spectrum and the calculated result. The overall lineshape and width is satisfactorily reproduced. In particular, a two-peak (T$_1$ and T$_2$) structure is observed at similar energies to the experimental M$_1$ and M$_2$ peaks, while the shoulder M$_1'$ near ~121 meV in experiment is also apparent in the calculated DOS (T$_1'$).

With reference to the magnon dispersions in Fig. 3a, M$_1$ and M$_2$ may be associated with two-magnon scattering involving the two highest energy branches at the Brillouin zone boundary, respectively. Meanwhile, the shoulder of M$_1$ at ~121 meV likely originates from two-magnon scattering involving the lowest energy branch at the Brillouin zone boundary. It should also be noted that the spin-wave interpretation given here differs from the discussion of the RIXS data presented in ref. [5], where the magnetic scattering was interpreted as a superposition of single site excitations.

Each term in the spin Hamiltonian ($J$, $D$ and $A$) affects the magnon DOS in a distinct manner, as shown in Supplementary Fig. 3. In combination with the high energy resolution of the Raman technique, these distinct effects allow us to accurately estimate the $J$, $D$, and $A$ parameters experimentally from the spectra. An increase of the $J$ parameter increases the overall separations and widths of the DOS peaks, reflecting changes in the bandwidths of the individual magnon branches, while leaving the onset in the DOS unchanged (Supplementary Fig. 3a, d). An increase in $D$ is observed to shift the peak positions and also reduce their separations (Supplementary Fig. 3b, e). Finally, an increase in $A$ is seen to increase the energies of T$_1'$, T$_1$ and T$_2$ equally, with the splitting between them remaining approximately constant (Supplementary Fig. 3c, f). Therefore, by comparing the calculated result to the dominant peak positions and onset in the data (Fig. 3c), we are able to estimate ($J$, $D$, $A$) = (5.1, 1.7, −5.3)

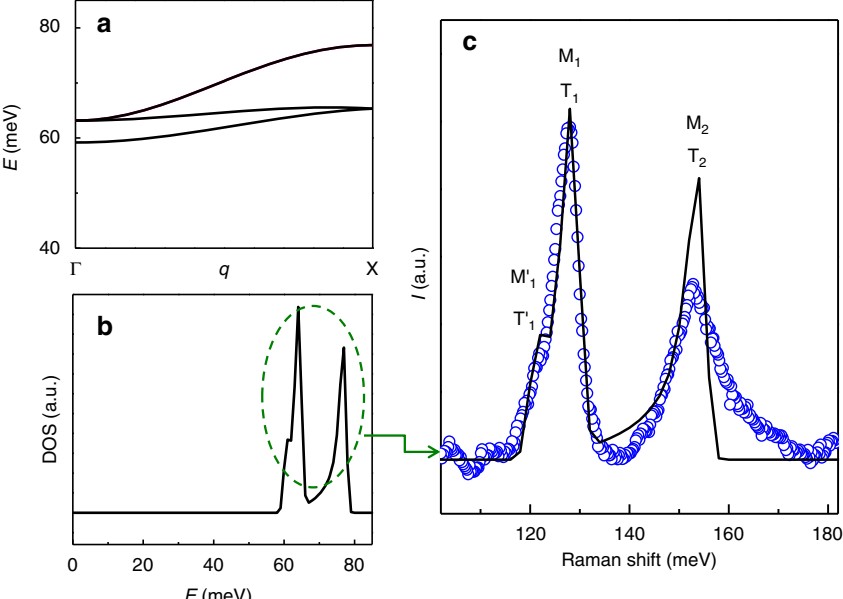

**Fig. 3** LSWT calculations for $Cd_2Os_2O_7$. **a** Spin wave dispersion showing three branches along the high symmetry Γ–X line for the AIAO ordered state. **b** Calculation of the magnon density of states DOS (a.u.). The spin wave and the magnon DOS are calculated with LSWT using values of $J = 5.1$ meV, $D = 1.7$ meV, $A = -5.3$ meV. **c** Comparison between the experimental spectrum of the two-magnon Raman scattering at 100 K (*open blue circles*) and the theoretical magnon DOS with energy scaled by factor of two (*black solid line*). The comparison confirms that the broad Raman peaks arise from two-magnon Raman scattering in the AIAO state, likely involving the two highest energy magnon branches at the X point

meV. These values differ considerably from the closest density functional theory estimates[2] ($J = 14$ meV, $D = 4$ meV, and $A = -24$ meV for $U_{eff} = 1.25$ eV), but are in excellent agreement with the many-body quantum-chemical results of Bogdanov et al.[7] ($J = 6.4$ meV, $D = 1.7$ meV, and $A = -6.8$ meV). The discrepancy between our LSWT estimates and the quantum-chemical results may be due to the neglect of final-state magnon–magnon interactions in our calculations.

**Evidence for spin-charge-lattice coupling.** $5d$ transition metal-oxides are characterized by a close coupling between spin, charge, and lattice degrees of freedom[26, 28]. To determine whether the spin excitations interact with other degrees of freedom in $Cd_2Os_2O_7$, we now turn to the phonon features present in the Raman data (Fig. 1a). All Raman modes reveal anomalies near $T_N$. However, we focus on the $T_{2g}(4)$ phonon near 96 meV, which shows the most pronounced anomalies as well as a distinct asymmetric (Fano) lineshape. Such a spectral lineshape can arise from the coupling of the phonon with a continuum of excitations and is often encountered in the Raman spectra of metals[29]. The ionic motion associated with this mode mostly involves O(1), the oxygen atom at the apex of the $OsO_6$ octahedron, and is illustrated in Supplementary Fig. 1f. As shown in Fig. 4a, the $T_{2g}(4)$ mode displays a pronounced Fano asymmetry at $T > T_N$ (i.e., 240 K), but becomes comparably symmetric at $T < T_N$ (i.e., 115 K). To quantify the observed changes, we have fitted the $T$-dependent spectra with the Breit–Wigner–Fano (BWF) profile[30]:

$$I(\omega) = I_0 \frac{[1 + (\omega - \omega_0)q\Gamma]^2}{1 + [(\omega - \omega_0)\Gamma]^2} \quad (2)$$

here $I_0$, $\omega_0$, $\Gamma$, and $1/|q|$ are the intensity, phonon frequency, width, and dimensionless asymmetry parameter of the BWF mode, respectively. As can be seen in Fig. 4a, the Fano fitting adequately describes the $T_{2g}(4)$ phonon both below and above $T_N$.

The $T$-dependence of the Fano parameters reveals anomalies near $T_N$ (i.e. at $T^* \sim 0.9 T_N$). Figure 4b shows that $1/|q|$ has a complex non-monotonic $T$-dependence. Above $T^*$, $1/|q|$ is relatively constant before dropping rapidly to a minimum near $0.8\, T_N$. It then grows as $T$ is reduced further. As shown in Fig. 4 (c), the $\omega$ and $\Gamma$ parameters show related anomalies near $T^*$: $\Gamma$ drops rapidly while $\omega_0$ reveals a rapid rise. Phonon modes usually show a smooth increase (decrease) in the frequency (width) at low temperatures due to lattice anharmonicity. The $T$-induced changes of $\omega_0$ and $\Gamma$ expected from anharmonicity can be evaluated from the anharmonic decay model[31] and are shown as the dashed lines in Fig. 4c. At low temperatures, the $\omega$ data show a clear kink at $T^*$ while the slope of $\Gamma$ changes abruptly below $T^*$, clearly deviating from the anharmonic model. Note that the occurrence of the phonon anomalies below $T_N$ is not an artifact due to local laser heating: the magnetic scattering intensity introduced in Fig. 2c clearly onsets at $T_N$.

The complex non-monotonic $T$-dependence of $1/|q|$ and the anomalies of $\Gamma$ below $T_N$ indicate coupling of the $T_{2g}(4)$ phonons to both charge and spin degrees of freedom. In the metallic region above $T_N$, $1/|q|$ remains constant and quite large, indicating that the $T_{2g}(4)$ phonon is strongly coupled with free carriers. At $T < T^*$ ($\sim 0.9 T_N$), $1/|q|$ is approximately proportional to $(T_N - T)$, implying the coupling to spin degrees of freedom as well. At $T^* < T < T_N$, the value of $1/|q|$ may be understood by combining these two coupling contributions. As shown in Fig. 4c, both the $\omega_0$ and $\Gamma$ parameters reveal anomalies at $T^*$, not at $T_N$. Such behavior suggests that there is a temperature region (i.e. $T^* < T < T_N$) where free carriers coexist with magnetism in $Cd_2Os_2O_7$, in agreement with a recent optical study[6].

**Discussion**

The important features of the magnetic scattering in $Cd_2Os_2O_7$ at $T \ll T_N$ can be explained using LSWT, implying the existence of well-defined dispersive magnons. Despite its overall success, however, the $T$-dependence and detailed lineshape of the two-

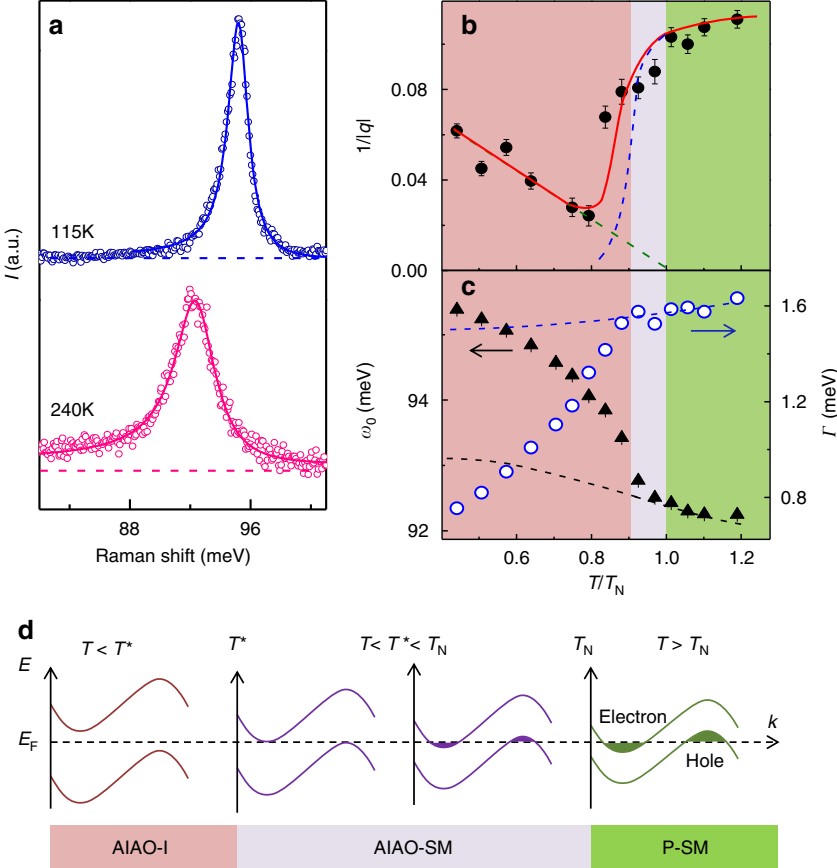

**Fig. 4** The anomalous lineshape and $T$-dependence of the $T_{2g}(4)$ phonon mode. **a** Raman spectra of the $T_{2g}(4)$ phonon mode taken at low (115 K) and high (240 K) temperature. The data (*open pink and blue circles*) show the Raman phonon peaks and are fitted with the Breit-Wigner-Fano function (*pink and blue solid lines*). The *dashed lines* indicate the frequency-independent background. **b** The $T$-dependence of the Fano asymmetry parameter. The experimental data are plotted as *black circles*. The *solid* and the *dashed lines* are guides for the eye. The *dashed lines* express the contribution of magnetic (*green dashed line*) and charge (*blue dashed line*) coupling to $1/|q|$, while the *red solid line* represents the total. **c** The frequency ($\omega_0$-*black triangles*) and the width ($\Gamma$-*white circles*) at various temperatures. The fitted anharmonic decay model is shown in *dotted lines*. Error bars in **b**, **c** represent the standard deviation in the data fitting procedure. **d** A schematic of the Liftshitz-type MIT, which present simplified band structures expected for the paramagnetic semimetal (*P-SM*), AIAO ordered semimetal (*AIAO-SM*), and AIAO ordered insulating (*AIAO-I*) states. Above $T_N$, the system is a P-SM with the same number of electrons and holes due to the even number of electrons in the unit cell. The filled *green* and *violet regions* represent the electron and hole pockets. The electron and hole bands overlap in energy, resulting in a semimetallic state. When $T$ decreases through $T_N$, the emergence of AIAO ordering shifts the electron and hole bands upwards and downwards respectively, gradually annihilating the Fermi surface. With a further decrease in $T$, the electron and hole pockets disappear and their overlap in energy eventually becomes zero at $T^*$. Below $T^*$, a true energy gap develops between the electron and hole bands, resulting in a robust insulating phase

magnon scattering reveal several unusual features that cannot be captured within LSWT. In terms of lineshape, we note that the $M_2$ peak has a clear tail extending to higher energies than expected from LSWT, which predicts a cutoff at ~160 meV (Fig. 3b, c). The discrepancies between the experimental and theoretical curves is probably due to the neglect of final-state magnon–magnon interactions in our calculations[32, 33], which are expected to be strong in the non-collinear AIAO magnetic state. In particular, the asymmetric DOS singularities seen in the calculation are expected to broaden, become more symmetric, and shift to lower energies once magnon–magnon interactions are properly considered. As can be seen in Fig. 2d, the frequency is renormalized by more than 15% at 180 K and the ratios $\Gamma_M(180\,K)/\Gamma_M(100\,K)$ is about 2.5. At $T_N$, we expect these values to reach 25% and 4, respectively. This behavior may also be attributed to magnon-magnon interaction effects, as these are expected to become more pronounced at elevated $T$ due to thermally-excited magnons. The strong $T$-dependence of the two-magnon peak energies and linewidths are in accord with the earlier reports of two-magnon scattering for three- dimensional

antiferromagnets, and much stronger than the two-dimensional (2D) case (where the frequency of the zone-boundary magnons are renormalized by less than 5% and $\Gamma$ becomes broadened by about a factor of 1.5 only)[22, 34].

An additional factor that may contribute to the above anomalies is Landau damping[8], which has been suggested in previous reports to be relevant to the AIAO state in the intermediate correlation regime[8, 11]. We note that the optical gap in our $Cd_2Os_2O_7$ crystals is only ~155 meV at low temperatures[6], slightly above the energy of the two-magnon scattering. The magnetic excitations may therefore overlap with the electron–hole continuum and Landau damping (electron–magnon interaction) can occur. As a result, the magnetic excitations may decay into electron–hole pair excitations, so the magnon lifetimes and energies may be renormalized. Consequently, Landau damping may contribute to the broadening and renormalization of the two-magnon peaks. As $T$ is increased towards $T_N$, the optical gap softens considerably[6]. Coupling with low-lying charge excitations, as well as magnon–magnon interaction effects, may therefore contribute to

the strong $T$-dependent damping and softening of $M_1$ and $M_2$ that is evident in Fig. 2a, as well as to the high-energy tail of the magnetic scattering discussed above.

The complete disappearance of the magnetic scattering above $T_N$ in $Cd_2Os_2O_7$ is also unusual (Figs 2a and 2c). This behavior should be contrasted with the $T$ evolution reported previously for two-magnon scattering in canonical insulating magnets. In three-dimensional systems, such as $MnF_2$ and $FeF_2$, the well-defined two-magnon scattering present at $T \ll T_N$ progressively softens, evolving into an intense quasi-elastic response at $T \gg T_N$[35–37]. Persistent scattering at $T > T_N$ is also observed in 2D systems, such as $HoMnO_3$[24], $Sr_2IrO_4$[26], and $K_2NiF_4$[22], although in these cases the magnetic scattering remains peaked at finite energies due to the presence of short-range 2D spin–spin correlations at experimental temperatures. Above $T_N$, $Cd_2Os_2O_7$ enters a metallic state, as evidenced by direct current transport and optical conductivity studies[4, 6, 38]. The disappearance of the magnetic scattering at $T_N$ may therefore be attributed to strong longitudinal spin fluctuations in the metallic state (in other words, the Os local moment itself is completely washed out)[39], in line with the exchange-enhanced Pauli-like susceptibility (e.g., non-Curie–Weiss susceptibility) reported by Mandrus et al. above $T_N$[38]. The complete suppression of magnetic scattering above $T_N$, as well as the comparable energies of the magnetic and charge excitations, is suggestive of itinerant magnetism which is valid in the limit of weakly interacting band electrons[40]. This should be contrasted with the success of LSWT, which presupposes well-defined local moments (and localized electrons), in describing the magnetic scattering at $T \ll T_N$. Taken together, these findings suggest that $Cd_2Os_2O_7$ manifests aspects of both localized and itinerant magnetism, and may therefore be placed in the intermediate electron correlation regime in the sense of ref. [11]. This is broadly consistent with the extended, and hence weakly-correlated, nature of the $5d$ Os orbitals[41].

Having described the magnetic excitations, we now focus on the anomalous temperature dependence of the $T_{2g}(4)$ phonon. The unusual $T$-dependence of the $T_{2g}(4)$ phonon in Fig. 4 can be understood in terms of the Lifshitz-type MIT, recently proposed for $Cd_2Os_2O_7$[2, 42]. It is depicted schematically in Fig. 4d. At $T > T_N$, the system is a paramagnetic semimetal with both electrons and hole bands at the Fermi level ($E_F$), leading to a large $1/|q|$ due to electron-phonon coupling. Below $T_N$, the system enters an AIAO-ordered semimetallic state. As the magnetic order develops, the electron and hole bands are repelled from $E_F$ and the electronic contribution to $1/|q|$ (the *dashed line* in Fig. 4b) is gradually reduced. At the crossover temperature $T^\star \sim 0.9 T_N$, the Fermi surface is entirely removed and $1/|q|$ rapidly drops towards its minimum value. Concurrently, $\Gamma$ is reduced and $\omega_0$ is renormalized due to the loss of the electron-hole decay channel (Fig. 4c). At still lower temperatures, the AIAO order continues to develop and the two-magnon scattering intensity (the tail of $M_1$ and $M_2$) increases (Fig. 2a). The magnetic continuum also couples to the $T_{2g}(4)$ phonon, causing $1/|q|$ to rise. Indeed, a linear approximation to the magnetic contribution to $1/|q|$ (*dashed-dot*) line in Fig. 4b extrapolates to zero near $T_N$, pointing to the magnetic origin of this component. Our estimate of $T^\star \sim 0.9 T_N \sim$ 204 K agrees well with the 210 K value estimated from the optical conductivity data[6]. Note that the coexistence of free carriers with AIAO magnetism also is predicted to occur in the pyrochlore iridate system, such as in the Weyl semimetal state[12, 43].

Our study has revealed that, in addition to the phonon modes reported in the literature[19], there exist broad peaks in the Raman response of $Cd_2Os_2O_7$ at higher energy. These broad peaks exhibit an anomalous behavior near the magnetic transition temperature and our analysis indicates that they originate from two-magnon scattering. The good agreement between the

observed two-magnon scattering peaks and the calculated spectral shape from LSWT establishes the existence of well-defined spin waves in the AIAO magnetic state of $Cd_2Os_2O_7$ and allows for a quantitative estimation of the effective spin Hamiltonian parameters. We explained certain differences between the magnetic scattering in $Cd_2Os_2O_7$ and in canonical insulating magnets by arguing that $Cd_2Os_2O_7$ should be placed in the intermediately-correlated regime.

Our results establish magnetic Raman scattering as a simple and powerful method for exploring the spin dynamics in $5d$ AIAO pyrochlore magnets, a task that has proven challenging with other techniques. For these systems, Raman scattering has two advantages. First, the high energy resolution of the Raman technique facilitates detecting fine structure in the magnetic excitation spectrum. Second, only a small sample mass is required (useful when only small single crystals or thin films are available). Indeed, pyrochlore iridate thin films are predicted to host topological phenomena and a significantly reduced local moment[44]. Raman scattering is well-positioned to provide insight into such emergent behavior by directly probing the associated magnetic excitations.

## Methods

**Synthesis and magnetization**. Single crystal $Cd_2Os_2O_7$ was grown by the chemical transport method. First, a polycrystalline pellet was prepared from a mixture including 5–10% excess CdO and Os in a sealed quartz tube, under supply of an appropriate amount of AgO as the oxygen source at 1073 K; addition of too much AgO should be avoided, as highly toxic $OsO_4$ might be produced. Second, the tube was kept in a furnace with a temperature gradient of 1040–1200 K for a week with the pellet at the high temperature side. Single crystals were then grown via a chemical vapor transport process. Magnetization was measured in a Quantum Design MPMS.

**Raman scattering**. The Raman scattering spectra of the $Cd_2Os_2O_7$ samples were obtained in the backscattering configuration with microprobe LabRAM HR Evolution (Horiba Co.) and NANOBASE XperRam 200 Raman spectrometers. For detection, we used a high-throughput single-stage spectrometer with a grating of 1800 lines per mm. The spectral resolutions are about 2 cm$^{-1}$ for the LabRAM and about 3 cm$^{-1}$ for the XperRam. Solid state lasers operating at 633 nm (1.96 eV) and 532 nm (2.33 eV) were used as the excitation sources. The excitation power and beam diameter on the sample surface were ~1 mW and ~50 μm, respectively. The sample was mounted in a compact temperature stage (THMS600, Linkam) for the $T$-dependent measurements from 100 to 270 K and a helium closed-cycle cryostat for 15 to 100 K.

**Linear spin wave theory**. The calculations of the magnon spectra have been performed using LSWT for the AIAO magnetic ground state assuming the Hamiltonian in Eq. (1).

In order to obtain the spin wave expansion about the noncollinear AIAO state, we first rewrite the Hamiltonian in Eq. (1) in terms of local spin variables $\tilde{S}_i = R_i^{-1} S_i$. Here $R_i$ is the rotation operator that takes the direction of magnetic ordering at site $i$ and rotates it to the $z$ axis of the global coordinate system. Next, the Hamiltonian is expressed in terms of bosonic operators $b_i$ and $b_i^+$ using the Holfstein-Primakoff transformation:

$$\tilde{S}_i^+ = \sqrt{2\bar{S}} b_i$$

$$\tilde{S}_i^- = \sqrt{2\bar{S}} b_i^+ \qquad (3)$$

$$\tilde{S}_i^z = S - b_i^+ b_i$$

where $\tilde{S}_i^\pm = \tilde{S}_i^x \pm i \tilde{S}_i^y$. Only the terms up to quadratic order in the bosonic operators are retained. Then, we Fourier transform the bosonic operators and obtain the excitation spectrum via a Bogoliubov transformation.

**Data availability**. The data that support the findings of this study are available from the corresponding author upon request.

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

## Acknowledgements

This work was supported by the Research Center Program of IBS (Institute for Basic Science) in Korea (IBS-R009-D1). S.J.M. acknowledges the support by Basic Science Research Program through the National Research Foundation of Korea (NRF) funded by the Ministry of Science, ICT, and Future Planning (2014R1A2A1A11054351 and 2017R1A2B4009413). I.-S.Y. acknowledges the support by the National Research Foundation of Korea (NRF) grant funded by the Korea government (MSIP) (No.2015001948). J.-H.K. acknowledges the support by the National Research Foundation of Korea (NRF) grant funded by the Korea government (MSIP) (NRF-2016R1A2B4012646). The work at Ames Laboratory was supported by the Critical Materials Institute, an Energy Innovation Hub funded by the U.S. Department of Energy (DOE). AmesLaboratory is operated for the U.S. DOE by Iowa State University under contract #DEAC02-07CH11358. T.M.H.N. thanks Dr Manh Cuong Nguyen from Ames Laboratory, U. S. DOE for helpful discussions.

## Author contributions

T.M.H.N. and T.W.N. conceived the investigation. T.M.H.N., I-S.Y., C.H.S., and J.-H.K. performed the Raman experiments and analyzed the data. Z.H. and J.Y. provided the single crystal sample and performed the magnetization experiment. A.L.W. performed the LSWT calculations. C.H.K. prepared the first-principles calculations. T.M.H.N., L.J.S., S.J.M., and T.W.N. wrote the manuscript. All co-authors contributed to discussion of the results.

## Additional information

**Competing interests:** The authors declare no competing financial interests.

**Change history:** A correction to this article has been published and is linked from the HTML version of this article.

