## [Peer Review File · Nature Communications]

Reviewers' comments:

Reviewer #1 (Remarks to the Author):

Nguyen Thi Minh Hien and coworkers have reported on the observation of two-magnon scattering in Cd₂Os₂O₇, a 5d pyrochlore system. Generally very little is known about magnetism in 5d materials due to either high neutron absorption or unfavorable sizes of the crystals. In this regard the discovery of two-magnon scattering in Cd₂Os₂O₇, through Raman measurements, is welcomed and offers us an independent way to determine the microscopic model of the system. Additionally Nguyen Thi Minh Hien and coworkers are able to look at the phonon modes and the coupling with charges and magnetism.

The work is convincing and well done. I support the publication of the manuscript if the following minor comments can be addressed.

1) Line 71: Please specify that the 25 meV resolution only applies to Ir. For Os I believe the best resolution is above 100 meV.

2) Figure 2b: Please use binning on the data. Far too many points for the broad two-magnon, making it appear somewhat noisy.

3) Based on optical spectroscopy (Ref 28) the optical gap gets fully filled above TN. In that case I would expect that the lower energy phonon modes would also show some anomalies. Can the author comment on why the same analysis, as for the T_{2g}(4) mode, was not applied to the remaining modes?

Reviewer #2 (Remarks to the Author):

In their manuscript Nguyen Thi Minh Hien et al. report on an experimental study of the magnetic excitations in Cd₂Os₂O₇ using Raman scattering.

The manuscript is well written, and the data are clearly presented and thoughtfully analysed.

Cd₂Os₂O₇ has been of enduring interest due to the fact that it exhibits a metal-insulator transition as a function of temperature. Claims have been made that the MIT in Cd₂Os₂O₇ can be classified in various ways, including as being paradigmatic of transitions associated with the names of Slater, Lifshitz, etc. The classification of the MIT in this material may indeed be considered to be an open question.

The claimed two main achievements of the paper are:

- (1) The extraction of detailed information on the magnetic interactions;
- (2) Identification of complex spin-charge-lattice coupling, supporting the notion that Cd₂Os₂O₇ is close to a Lifshitz instability.

The claims are probably well supported by the data presented and their analysis.

Of the two, the second is more interesting and potentially profound - the first may be considered to be consistent with earlier quantum chemistry calculations and RIXS experiments.

However, overall I consider that the manuscript represents significant progress in our understanding of Cd₂Os₂O₇, and satisfies the criteria for publication in Nature Communications.

Specific comments that need addressing before the paper can be accepted:

Line 149

A number of issues need to be resolved here and elsewhere.

The values of the parameters cited are in fact more or less the same as those provided by the quantum chemistry calculations. This should be stated here, not later.

Please explain how the parameters were obtained? Least squares fit?

Also, please provide error bars on the parameters, here and everywhere in the manuscript where they are stated.

Line 175

"Each term in the spin Hamiltonian (A, J and D) affects the magnon DOS in a distinct fashion."

Please provide details of the covariance matrix which will reveal the degree of independence of the parameters.

Line 528:

I am somewhat puzzled by the results of the LSWT presented in Fig. 3.

In particular, given the large values of D and A I am surprised that one of the spin-wave modes exists at the low energies shown.

Could the authors please double check their calculations as

this is a crucial point. I could not reproduce this mode

in calculations performed using standard software such as SpinW, but admit that the error may be in my calculations.

Reviewer #3 (Remarks to the Author):

This manuscript presents an experimental Raman scattering study of AIAO state in Cd₂Os₂O₇. The magnetic degrees of freedom in this compound are spins-3/2 which interact via anisotropic interactions, among which single-ion anisotropy and Heisenberg interaction are very similar in magnitude.

The authors observed two peaks at 125 meV and 145 meV and provided experimental evidences that these two peaks are of the magnetic origin. This is a very nice but rather well-expected result. Indeed, the authors managed to describe the observed Raman spectra with a very standard approach based on the LSW calculation, and based on this they claim that the magnon-magnon interactions are not important. This statement is probably arguable, because both the effects of the final-state interactions and the renormalization of the magnon spectra due to these interactions are expected to be important in the non-collinear AIAO state even for S=3/2. Also, I think that there is a contradiction in the fact that the linewidth is strong temperature dependent but the magnon-magnon interactions are not important.

The authors suggested a possible explanation based on the Landau damping but this proposal was not really elaborated in the manuscript. I was also surprised that the authors did not discuss the polarization dependence of the Raman response since it could provide important additional information.

The authors also observed that the Raman response disappears above the Neel temperature. The authors suggest that the system might be almost metallic above certain temperature and, therefore, exhibit non Curie-Weiss behavior. I believe that this statement can not be justified based solely on the Raman data and temperature dependence of some transport properties should be done in order to confirm it.

Overall, I believe this manuscript presents a solid research which can be, in principle, improved by addressing the questions raised above. However, I do not find this work in particular novel and deserving the publication in Nature Communication. At least, in the current version of the manuscript the authors did not stress it firmly enough.

I. Response to Referee #1 comments

Nguyen Thi Minh Hien and coworkers have reported on the observation of two-magnon scattering in $\text{Cd}_2\text{Os}_2\text{O}_7$, a $5d$ pyrochlore system. Generally very little is known about magnetism in $5d$ materials due to either high neutron absorption or unfavorable sizes of the crystals. In this regard the discovery of two-magnon scattering in $\text{Cd}_2\text{Os}_2\text{O}_7$, through Raman measurements, is welcomed and offers us an independent way to determine the microscopic model of the system. Additionally Nguyen Thi Minh Hien and coworkers are able to look at the phonon modes and the coupling with charges and magnetism.

The work is convincing and well done. I support the publication of the manuscript if the following minor comments can be addressed.

1) Line 71: Please specify that the 25 meV resolution only applies to Ir. For Os I believe the best resolution is above 100 meV.

We thank the referee for pointing this out. On page 3 in the revised manuscript, we now write:

“The resolution of state of the art RIXS is still only 25 meV for Ir⁸ and above 40 meV for Os¹⁴,”

2) Figure 2b: Please use binning on the data. Far too many points for the broad two-magnon, making it appears somewhat noisy.

Since the magnon peaks are broad, we could have done the Raman measurements with high number of x-binning in order to increase signal-to-noise ratio of the weak magnon peaks. On the other hand, we wanted to keep the spectroscopic resolution for studying of phonon modes. Therefore, in the revised figure, we apply a “5-point” average to the two-magnon data in Fig. 2b to reduce the noise, but leave the data unmodified elsewhere to avoid affecting the lineshapes of the narrow phonon peaks.

3) Based on optical spectroscopy (Ref 28) the optical gap gets fully filled above T_N . In that case I would expect that the lower energy phonon modes would also show some anomalies.

Can the author comment on why the same analysis, as for the $T_{2g}(4)$ mode, was not applied to the remaining modes?

The referee is correct that the lower energy modes also show anomalies at T_N . However, the effect is much stronger in the $T_{2g}(4)$ mode, so we focus our analysis on this feature. The details of the phonon anomalies will be presented in a later paper.

[Redacted]

Response to Referee #2 comments

In their manuscript Nguyen Thi Minh Hien *et al.* report on an experimental study of the magnetic excitations in $\text{Cd}_2\text{Os}_2\text{O}_7$ using Raman scattering. The manuscript is well written, and the data are clearly presented and thoughtfully analysed.

$\text{Cd}_2\text{Os}_2\text{O}_7$ has been of enduring interest due to the fact that it exhibits a metal-insulator transition as a function of temperature. Claims have been made that the MIT in $\text{Cd}_2\text{Os}_2\text{O}_7$ can be classified in various ways, including as being paradigmatic of transitions associated with the names of Slater, Lifshitz, etc. The classification of the MIT in this material may indeed be considered to be an open question.

The claimed two main achievements of the paper are:

- (1) The extraction of detailed information on the magnetic interactions;
- (2) Identification of complex spin-charge-lattice coupling, supporting the notion that $\text{Cd}_2\text{Os}_2\text{O}_7$ is close to a Lifshitz instability. The claims are probably well supported by the data presented and their analysis.

Of the two, the second is more interesting and potentially profound - the first may be considered to be consistent with earlier quantum chemistry calculations and RIXS experiments.

However, overall I consider that the manuscript represents significant progress in our understanding of $\text{Cd}_2\text{Os}_2\text{O}_7$, and satisfies the criteria for publication in Nature Communications.

Specific comments that need addressing before the paper can be accepted:

1) Line 149: A number of issues need to be resolved here and elsewhere. The values of the parameters cited are in fact more or less the same as those provided by the quantum chemistry calculations. This should be stated here, not later. Please explain how the parameters were obtained? Least squares fit? Also, please provide error bars on the parameters, here and everywhere in the manuscript where they are stated.

Final state magnon-magnon interactions (see reply to referee #3) and matrix element effects make a direct least-squares fit of the Raman data difficult. In our work, the parameters were simply chosen by matching by eye the peak positions in the calculated DOS with the experimental peak positions. In our work, we initially used the parameters which were reported in the many body QM calculations (ref 7). Then we tuned them to obtain better agreement with the data. We could obtain a decent agreement when we reduced anisotropy A value by 22% and exchange integer J value by 20%. In the Supplementary, we explain in detail about the effects of each term in the spin Hamiltonian on the magnon DOS and our procedure for estimating them.

We describe our method explicitly on page 7 in the revised manuscript where we write:

“Therefore, by comparing the calculated result to the dominant peak positions and onset in the data (Fig. 3c), we are able to estimate $(J, D, A) = (5.1, 1.7, -5.3) \text{ meV}$.”

With regards to the quantum chemistry results, on page 6 in the revised manuscript we now write:

“For J , D and A , we used values of 5.1 meV, 1.7 meV, and -5.3 meV, respectively. As discussed in detail later, these values yield good agreement with the Raman data and also closely agree with the results of many-body quantum chemical calculations.”

2) Line 175 : "Each term in the spin Hamiltonian (A , J and D) affects the magnon DOS in a distinct fashion." Please provide details of the covariance matrix which will reveal the degree of independence of the parameters.

As mentioned above, we did not perform a least squares fit, so no covariance matrix was generated.

3) Line 528: I am somewhat puzzled by the results of the LSWT presented in Fig. 3. In particular, given the large values of D and A , I am surprised that one of the spin-wave modes exists at the low energies shown. Could the authors please double check their calculations as this is a crucial point. I could not reproduce this mode in calculations performed using standard software such as Spin W, but admit that the error may be in my calculations.

We are deeply thankful to Referee #2 for pointing out this issue. Following his/her comments, we checked our calculations carefully. Unfortunately, we found that we made a mistake in our original calculation, namely, we put the wrong signs in some of the DM vectors. This problem was fixed, and we redid the LSWT calculations again for the revised manuscript. We very much appreciate Referee #2's detailed scrutiny of our work.

In the revised calculation, the lower spin wave branch is indeed pushed to higher energy. Actually, this improves the agreement with our data, as no magnetic scattering is resolved at low T in the low-energy range.

II. Response to Referee #3 comments

This manuscript presents an experimental Raman scattering study of AIAO state in $\text{Cd}_2\text{Os}_2\text{O}_7$. The magnetic degrees of freedom in this compound are spins-3/2 which interact via anisotropic interactions, among which single-ion anisotropy and Heisenberg interaction are very similar in magnitude.

1) The authors observed two peaks at 125 meV and 145 meV and provided experimental evidences that these two peaks are of the magnetic origin. This is a very nice but rather well-expected result. Indeed, the authors managed to describe the observed Raman spectra with a very standard approach based on the LSW calculation, and based on this they claim that the magnon-magnon interactions are not important. This statement is probably arguable, because both the effects of the final-state interactions and the renormalization of the magnon spectra due to these interactions are expected to be important in the non-collinear AIAO state even for $S=3/2$. Also, I think that there is a contradiction in the fact that the linewidth is strong temperature dependent but the magnon-magnon interactions are not important.

In our initial submission, we argued against magnon-magnon interaction effects due to the good agreement between the LSWT and our data. However, as mentioned in our reply to Referee # 2, we corrected an error in our LSWT calculations. Although the calculations still provide a good description of the data overall, certain aspects (especially the lineshapes of the two-magnon peaks compared with the revised LSWT calculation) are consistent with final state magnon-magnon interaction effects. Following the referee's suggestion, we tried to include the magnon-magnon interactions. In the revised manuscript, we just point out that the asymmetric DOS singularities seen in the calculation can be expected to shift to lower energy, broaden, and become more symmetric due to magnon-magnon interactions. The highest energy two-magnon peak is indeed more symmetric compared with the LSWT result. Furthermore, we note that our estimated J and A values are reduced by $\sim 20\%$ vs. the quantum chemical calculations of Ref. 14. This may be naturally explained by final state magnon-magnon interactions. We therefore agree with Referee # 3 that magnon-magnon effects are likely important in this compound. A complete theory of two-magnon Raman scattering in $\text{Cd}_2\text{Os}_2\text{O}_7$ including magnon-magnon interactions is beyond the scope of the presentwork but would be an interesting direction for further study.

On page 8 and page 9 in the revised manuscript, we now write:

“The discrepancies between the experimental and theoretical curves is probably due to the neglect of final state magnon-magnon interactions in our calculations^{32,33}, which are expected to be strong in the non-collinear AIAO magnetic state. In particular, the asymmetric DOS singularities seen in the calculation are expected to broaden, become

more symmetric, and shift to lower energies once magnon-magnon interactions are properly considered. As can be seen in Fig. 2d, the frequency is renormalized by more than 15% at 180K and the ratios $\Gamma_M(180K) / \Gamma_M(100K)$ is about 2.5. At T_N , we expect these values to reach 25% and 4, respectively. This behavior may also be attributed to magnon-magnon interaction effects, as these are expected to become more pronounced at elevated T due to thermally-excited magnons. The strong T -dependence of the two-magnon peak energies and linewidths are in accord with the earlier reports of two-magnon line shape for three-dimensional antiferromagnets, and much stronger than the two-dimensional case (where the frequency of zone-boundary magnon are renormalized by less than 5% and Γ becomes broadened by about a factor of only 1.5)^{22,34}.”

2) The authors suggested a possible explanation based on the Landau damping but this proposal was not really elaborated in the manuscript

We have edited the text to hopefully better explain this point. On page 9, we now write:

“An additional factor that may contribute to the above anomalies is Landau damping⁸, which has been suggested in the previous reports to be relevant to the AIAO state in the intermediate-correlation regime^{8,11}. We note that the optical gap in our $\text{Cd}_2\text{Os}_2\text{O}_7$ crystals is only ~ 155 meV at low temperatures⁶, slightly above the energy of the two-magnon scattering. The magnetic excitations may therefore overlap with the electron-hole continuum and Landau damping (electron-magnon interactions) can occur. As a result, the magnetic excitations may decay into electron-hole pair excitations, so the magnons lifetime increases and its peak position also changes. Consequently, Landau damping may contribute to the broadening and renormalization of the two-magnon peaks. As T is increased towards T_N , the optical gap softens considerably⁶. Coupling with low-lying charge excitations, as well as magnon-magnon interaction effects, may therefore contribute to the strong T -dependent damping and softening of M_1 and M_2 peaks that is evident in Fig. 2a, as well as to the high energy tail of the magnetic scattering discussed above.”

I was also surprised that the authors did not discuss the polarization dependence of the Raman response since it could provide important additional information.

We have added Fig. S2 to the Supplementary Material which shows the polarization dependence. On page 4 in the revised manuscript, we now write:

“The polarization properties of M_1 and M_2 were investigated in three input-output polarization geometries: collinear, crossed, and unpolarized. We found that the lineshape and Raman shift of M_1 and M_2 are independent of polarization geometry (Supplementary Fig 2). For the remainder of our discussion, we focus on the unpolarized case, as the overall intensity is larger in this geometry.”

3) The authors also observed that the Raman response disappears above the Néel temperature. The authors suggest that the system might be almost metallic above certain temperature and, therefore, exhibit non Curie-Weiss behavior. I believe that this statement cannot be justified based solely on the Raman data and temperature dependence of some transport properties should be done in order to confirm it.

The metallic state of $\text{Cd}_2\text{Os}_2\text{O}_7$ above T_N is well-established by, for example, DC transport and optical conductivity studies^{4,6}. Mandrus *et al*³⁸ also showed that the susceptibility in the metallic state is non-Curie-Weiss-like.

On page 10 in the revised manuscript, we now write:

*“Above T_N , $\text{Cd}_2\text{Os}_2\text{O}_7$ enters a metallic state, as evidenced by DC transport and optical conductivity studies^{4,6,38}. The disappearance of the magnetic scattering at T_N may therefore be attributed to strong longitudinal spin fluctuations in the metallic state (in other words, the Os local moment itself is completely washed out)³⁹, in line with the exchange-enhanced Pauli-like susceptibility (e.g., non-Curie-Weiss susceptibility) reported by Mandrus *et al* above T_N ³⁸.”*

Overall, I believe this manuscript presents a solid research which can be, in principle, improved by addressing the questions raised above. However, I do not find this work in particular novel and deserving the publication in Nature Communication. At least, in the current version of the manuscript the authors did not stress it firmly enough.

Our manuscript merits publication for several reasons. It contains new insight into the intriguing physics of $\text{Cd}_2\text{Os}_2\text{O}_7$ and provides the best experimental estimates to date of the magnetic Hamiltonian. More broadly, our study represents the first observation (to the best of our knowledge) of two-magnon Raman scattering in a $5d$ pyrochlore antiferromagnet and will motivate future studies of related materials, in particular

pyrochlore iridates. We would also like to point out that the other two referees agree with our assessment.

To clearly communicate the novelty of our findings, we have revised our introduction to read:

“Our results constitute the first detection of two-magnon excitations in a 5d pyrochlore oxide using Raman scattering. Through a comparison with linear spin wave theory (LSWT), we make quantitative estimates of the magnetic couplings that are in good agreement with recent many body quantum-chemical calculations⁷ and clarify the impact of the DMI, SIA, and Heisenberg-exchange interaction on the magnetic excitation spectrum. Understanding the nature of the magnetic excitations and the role of the microscopic magnetic interactions provides a foundation for deeper and fundamental discussion regarding the magnetic properties. Our measurements also reveal signatures of non-trivial coupling between spin, charge and lattice degrees of freedom across the magnetically-driven MIT which indicates that this transition may be classified as Lifshitz-type. These results provide new insight into the interplay between electronic structure and magnetism in 5d pyrochlore magnets and highlight the utility of Raman scattering for exploring the spin dynamics of other 5d AIAO pyrochlore magnets, such as the closely related pyrochlore iridates.”

REVIEWERS' COMMENTS:

Reviewer #1 (Remarks to the Author):

I thank the authors for addressing my questions and comments. The manuscript has been improved and a separate paper has been published focusing on the anomalies in the other phonon modes. In my opinion, this manuscript deserves a publication in Nature Communications.

Reviewer #3 (Remarks to the Author):

I am very happy with the authors' reply to my question and to the questions by other reviewers. I am also satisfied with their modification of the manuscript. Now I can recommend the paper for publication in Nature Communications.

REVIEWERS' COMMENTS:

Reviewer #1 (Remarks to the Author):

I thank the authors for addressing my questions and comments. The manuscript has been improved and a separate paper has been published focusing on the anomalies in the other phonon modes. In my opinion this manuscript deserves a publication in Nature Communications.

Reviewer #3 (Remarks to the Author):

I am very happy with the authors' reply to my question and to the questions by other reviewers. I am also satisfied with their modification of the manuscript. Now I can recommend the paper for publication in Nature Communications.

Our response:

We would like to thank the referees for appreciating our work. We also appreciate the referees for carefully reading our manuscript and for giving valuable comments which substantially helped improving the quality of the paper.